# Healthcare Resource Consumption and Related Costs in Patients on Antiretroviral Therapies: Findings from Real-World Data in Italy

**DOI:** 10.3390/ijerph20053789

**Published:** 2023-02-21

**Authors:** Valentina Perrone, Melania Dovizio, Diego Sangiorgi, Margherita Andretta, Fausto Bartolini, Arturo Cavaliere, Andrea Ciaccia, Alessandro Chinellato, Alberto Costantini, Stefania Dell’Orco, Fulvio Ferrante, Simona Gentile, Antonella Lavalle, Rossella Moscogiuri, Elena Mosele, Cataldo Procacci, Davide Re, Fiorenzo Santoleri, Alessandro Roccia, Franco Maggiolo, Luca Degli Esposti

**Affiliations:** 1CliCon S.r.l. Società Benefit—Health, Economics & Outcomes Research, 40137 Bologna, Italy; 2Azienda ULSS 8 Berica, 36100 Vicenza, Italy; 3USL Umbria 2 Terni, 05100 Terni, Italy; 4ASL Viterbo, 01100 Viterbo, Italy; 5Servizio Farmaceutico Territoriale ASL Foggia, 71121 Foggia, Italy; 6Azienda ULSS 3 Serenissima, 30174 Mestre, Italy; 7ASL Pescara, 65100 Pescara, Italy; 8ASL Roma 6, 00041 Albano Laziale, Italy; 9ASL Frosinone, 03100 Frosinone, Italy; 10Direzione Generale per la Salute Regione Molise, 86100 Campobasso, Italy; 11ASL Taranto, 74121 Taranto, Italy; 12UOC Assistenza Farmaceutica Territoriale, Azienda ULSS 7 Pedemontana, 36061 Bassano del Grappa, Italy; 13Dipartimento Farmaceutico ASL BAT, 76125 Trani, Italy; 14ASL Teramo, 64100 Teramo, Italy; 15Gilead Sciences S.r.l., 20124 Milano, Italy; 16ASST Papa Giovanni XXIII, 24127 Bergamo, Italy

**Keywords:** antiretroviral therapies (ART), human immunodeficiency virus (HIV), tenofovir alafenamide (TAF)-based regimens, drug utilization, adherence, persistence, healthcare resource consumption, direct healthcare costs, real-word evidence

## Abstract

This real-world analysis conducted on administrative databases of a sample of Italian healthcare entities was aimed at describing the role of therapeutic pathways and drug utilization in terms of adherence, persistence, and therapy discontinuation in HIV-infected patients under antiretroviral therapies (ART) and Tenofovir Alafenamide (TAF)-based regimens on healthcare resource consumption and related direct healthcare costs. Between 2015 and 2019, adults (≥18 years) prescribed with TAF-based therapies were identified and characterized in the year prior to the first prescription (index-date) for TAF-based therapies and followed-up until the end of data availability. Overall, 2658 ART-treated patients were included, 1198 of which were under a TAF-based regimen. TAF-based therapies were associated with elevated percentages of adherence (83.3% patients with proportion of days covered, PDC > 95% and 90.6% with PDC > 85%) and persistence (78.5%). The discontinuation rate was low in TAF-treated patients, ranging from 3.3% in TAF-switchers to 5% in naïve. Persistent patients had lower overall mean annual healthcare expenditures (EUR 11,106 in persistent vs. EUR 12,380 in non-persistent, *p* = 0.005), and this trend was statistically significant also for costs related to HIV hospitalizations. These findings suggest that a better therapeutic management of HIV infection might result in positive clinical and economic outcomes.

## 1. Introduction

The development and introduction of antiretroviral therapies (ART) and then the availability of combination ART in the clinical practice has dramatically changed the course of human immunodeficiency virus (HIV) infection from an acute to a chronic disease [1]. Nowadays, life expectancy for people living with HIV (PLWH) is up to that of the general population in those patients who receive a prompt diagnosis and early treatment and achieve adequate viral-load suppression [2]. International guidelines strongly recommend initiating an ART combination soon after diagnosis in accordance with patient needs, attitudes, and readiness [3,4]. The main goal of ART is to inhibit viral replication, to control the progression of HIV-infection into AIDS, to improve the overall clinical outcomes, to prevent the development of antiretroviral resistance, and to reduce HIV transmission [5].

According to the World Health Organization (WHO), adherence is defined as “the extent to which a person’s behaviour—taking medication, following a diet, and/or executing lifestyle changes, corresponds with agreed recommendations from a health care provider” [6]. Adherence to ART plays a key role in maximizing the viral suppression. Historically, the threshold for adherence was set at ≥95% based on studies conducted on the early era of ART [7,8]. With the second generation of ART, the cut-off level of adherence needed to achieve HIV viral suppression is shifting to lower values such as 80–85% that still allow to achieve a successful immunosuppression [7,9,10,11]. Nevertheless, long-lasting worldwide experience has shown that maintaining an optimal level of adherence to ART is still an unsolved clinical problem [12]. Thus, improving adherence to ART represents an important goal for both patients’ outcomes and healthcare savings in terms of resource utilization and related costs [13]. Among the most widely used ART combinations for PLWH, the Tenofovir Alafenamide (TAF)-based regimen has been shown to achieve a successful viral load suppression, but the results can be influenced by drug administration schedule. Complex regimens often remain among the few therapeutic options for PLWH with previous ART failure or resistance, but they are associated with an increased rate of adverse events and drug–drug interactions, ultimately leading to a higher risk of discontinuation. Conversely, ART simplification could lead to improved adherence levels, lower discontinuation rates, and higher patient satisfaction, feasibly due to the advantage of taking a single pill [14]. Therapeutic persistence, defined as the time that a patient remains on ART before interrupting or switching to another regimen, is also a crucial factor to ensure successful management of HIV, and the discontinuation of therapy has been related not only to lack of sustained viral suppression but also to toxicity or simplification independently from the level of adherence [15,16]. The progress made with HIV therapy is reflected on the shifting paradigm of the economic burden of HIV care, which was previously due mostly to hospitalization and management of opportunistic illness. The introduction of potent combinations of ART that elongated the life expectancy of PLWH made a substantial contribution in decreasing HIV-related hospitalizations and made HIV one of the most expensive chronic diseases, driven mostly by drug costs [17]. 

This real-world analysis was aimed at describing the demographic and clinical characteristics of patients treated with ART and evaluating therapeutic pathways and drug utilization. A special focus was placed on adherence, persistence, and therapy discontinuation of Tenofovir Alafenamide (TAF)-based regimens either as multi-tablet regimens (MTR) or single-tablet regimens (STR) among naïve patients and those with previous ART who switched for the first time to TAF-based regimens and the resulting impact on healthcare resource consumptions and related direct costs for the Italian National Health Service (INHS).

## 2. Materials and Methods

### 2.1. Data Source

The present retrospective observational analysis was based on data collected from the administrative databases of a sample of Italian Local Health Units (LHUs), covering around 5.52 million health-assisted subjects belonging to Apulia, Abruzzo, Molise, Campania, Lazio, Umbria, Liguria, Piedmont, and Veneto regions. The following databases were used for the analysis: (1) demographic database to obtain data on age, gender, and mortality; (2) pharmaceutical database for information related to drug prescriptions, such as anatomical-therapeutic chemical (ATC) code, number of packages, number of units per package, costs, and prescription date; (3) hospitalization database, including all hospitalization data with discharge diagnosis codes classified according to the International Classification of Diseases, Ninth Revision, Clinical Modification (ICD-9-CM), diagnosis-related group (DRG), and DRG-related charge; and (4) outpatient specialist services database, reporting data on prescriptions, type, description of diagnostic tests, and specialist visits. 

To guarantee patients’ privacy, an anonymous univocal numeric code (patient ID) was assigned to each health-assisted subject by the LHUs. The patient ID code allowed the electronic linkage between the databases. The anonymous code of the patient ensures the anonymity of the extracted data in full compliance with UE Data Privacy Regulation 2016/679 (“GDPR”) and Italian D.lgs. n. 196/2003, as amended by D.lgs. n. 101/2018. All the results of the analyses were produced as aggregated summaries, which could not be connected, either directly or indirectly, to individual patients. 

### 2.2. Design of the Analysis

#### Population Included

All adult (≥18 years old) patients prescribed with ART between January 2015 and December 2019 were screened. Among them, patients with TAF-based therapies were identified during all inclusion period by presence of the following drugs: TAF (ATC code J05AF13); emtricitabine and tenofovir alafenamide (ATC code J05AR17); emtricitabine, tenofovir alafenamide, elvitegravir, and cobicistat (ATC code J05AR18); emtricitabine, tenofovir alafenamide, and rilpivirine (ATC code J05AR19); emtricitabine, tenofovir alafenamide, and bictegravir (ATC code J05AR20); and emtricitabine, tenofovir alafenamide, darunavir, and cobicistat (ATC code J05AR22). Patients with TAF-based therapies were defined as naïve if they did not present any ART prescription in the 12 months before starting TAF-based therapies and as TAF-switchers if they had switched from a non-TAF-based to a TAF-based regimen.

Patients who moved to another LHU during the period considered in the analysis or with just one ART prescription or with prescription gaps greater than 12 months between subsequent prescriptions were excluded from the analysis. The date of the first prescription for TAF detected within the inclusion period was defined as the index date. Patients’ clinical characteristics were evaluated during at least 12 months prior to the index date (characterization period), while patients were followed-up until the end of database availability, which was at least 12 months for each patient. 

### 2.3. Variables Used in the Analysis

At baseline, the following comorbidities were evaluated: depression, respiratory disease, renal failure, alcohol/drugs abuse, cardiovascular disease, neoplastic diseases, diabetes, dyslipidemia, HBV/HCV, and hypertension (codes for identification are reported in Appendix A). Comorbidity profile was scored using a modified version of the Charlson Comorbidity Index (CCI) not accounting for HIV [18].

Drug utilization of TAF-based regimens was evaluated during follow-up. Adherence was measured as proportion of days covered (PDC), namely the ratio between the number of days of medication supplied and days of therapy, multiplied by 100 [19]. To define adherent patients, two thresholds were selected: the most conservative one, i.e., PDC > 95%, and a PDC > 85%, which represents a newly hypothesized threshold for adherence in view of novel ART entering the market [8]. Persistence to TAF-based regimens was defined as presence of TAF-based regimens prescriptions or any ART in the last quarter of 1-year follow-up. Discontinuation was defined as absence of any ART in the last quarter of follow-up. Switch was identified as a change from TAF-based to a not-TAF-based regimen during follow-up. Changes to a different TAF-based scheme or from TAF-MTR to TAF-STR were not considered a switch of therapy. The analysis of healthcare resource consumptions was reported as mean annual number of the following items for each alive patient: the mean annual number of prescription for HIV-related drugs, mean annual number of other medications, mean annual number of HIV-related hospitalization (DRG 488, 489, 490), mean annual number of hospitalizations for other causes, and mean annual number of outpatient specialist services (visits/diagnostic tests). The related direct healthcare costs in euros (EUR) sustained by INHS were retrieved from the administrative flows of the LHUs involved, which collect data for reimbursement purposes. The mean annual healthcare costs per patient were calculated as overall mean annual costs and divided by cost item, namely drug treatment (HIV-related drugs and other drugs), hospital admissions (HIV-related hospitalization and other hospitalizations), and outpatient specialist services. Healthcare resource consumptions and costs were stratified by persistence and level of adherence. Outliers (patients for whom total cost exceeds the mean cost plus 3-fold standard deviation) were excluded from cost analysis. Drug costs were calculated based on the purchase price by INHS. Hospitalization costs were determined using DRG tariffs. The costs of outpatient services (visits/tests) were defined according to tariffs applied by each region.

### 2.4. Statistical Analysis

Continuous variables are reported as mean ± standard deviation and categorical variables as numbers and percentages. A generalized linear model (GLM) with gamma distribution and identity link function was developed to assess the relationship between adherence level and non-ART costs. Statistical significance was accepted at *p* < 0.05. All the analyses were performed using Stata SE version 12.0 (StataCorp, College Station, TX, USA).

## 3. Results

### Overall ART-Treated Population 

A total of 2658 adult patients treated with ART were included in the analysis, and their demographic and clinical characteristics in the overall population and by calendar year are shown in Table 1.

In the total inclusion period, the mean age was 48.6 years, with the majority of patients in the age range of 36–55 years (61.6%). Male gender was more represented (71.6%). Analyzing the demographic characteristics by each year of inclusion, the male gender remained more frequent (71–73%), and a slight increase in age was noticed over the years. In the overall inclusion period, the most frequent comorbidities were hypertension (19.5%), followed by respiratory disease (15.9%) and dyslipidemia (11.2%) and mild average CCI (0.2) with 87.2% of patients with CCI score 0. The pattern of comorbidity distribution analyzed by each calendar year mirrored that of the total inclusion period. 

#### Patients Treated with TAF-Based Regimens

The analysis focused on 1198 patients prescribed a TAF-based regimen during inclusion period. Table 2 provides the demographic and clinical characteristics of overall TAF-treated patients and in subgroups divided into MTR (N = 514, 43%), STR (N = 684, 57%), naïve (N = 478, 40%), and TAF-switchers (N = 720, 60%). 

Adherence and persistence in the subgroups and in the total sample population were then investigated. In detail, the proportion of patients adherent to TAF-based regimens was 83.3% with a PDC threshold above >95%; specifically, it was higher among TAF-switchers (85.6%), followed by TAF-STR (84.4%), TAF-MTR (81.9%), and naïve (79.9%). The proportion of adherent patients with PDC > 85% was 90.6% (92.8% among TAF-switchers, 90.9% in TAF-STR, 90.1% in TAF-MTR, and 87.2% in naïve). Data on persistence revealed that the proportion of patients persistent to TAF-based regimens was 78.5% and reached 96% when persistence to any ART was analyzed. Switch to another ART not TAF-based occurred in 18% of patients, and 4% discontinued the therapy. The stratification in subgroups confirmed in general this overall trend, in particular for the rate of discontinuation that ranged between 3.3% (TAF-switchers) and 5% (naïve). Of note, patients under MRT regimen compared to those with STR showed a greater tendency to switch (30.4% vs. 8.8%, respectively) and a lower persistence on TAF-based therapy (66.5% vs. 87.4%, respectively). Overall persistence for TAF naïve patients was 74.3%.

The healthcare resource consumption during first year of follow-up for TAF-based regimes patients evaluated in the last 3 years of inclusion (2017–2018–2019) showed the highest numbers for ART prescriptions and specialistic visits/diagnostic tests, followed by other drug prescriptions, with a decreasing trend over time (Appendix A). The related mean annual total costs per patient are consistent with the data of healthcare resource consumption (Figure 1). A decreasing trend over the years was observed, with mean annual total costs per patients of EUR 15,493 for patients included in 2017, EUR 12,060 for those included in 2018, and EUR 10,216 for those included in 2019, mostly burdened by ART expenses.

The evaluation of the role of persistence on healthcare resources consumptions during the first year of follow-up in patients with TAF-based regimen revealed that the most relevant consumptions were attributable to ART prescriptions and specialistic visits/diagnostic tests and, lastly, other drug prescriptions to a higher extent in non-persistent patients (Appendix A). Regarding the cost analysis shown in Figure 2, patients persistent to TAF-based therapies had significantly lower mean annual costs than the non-persistent ones (EUR 11,106 vs. EUR 12,380, *p* = 0.005), and this difference was mostly appreciable in the cost item related to HIV hospitalization. Moreover, when the analysis was focused only on patients with HIV-related hospitalization, the mean costs were found to be higher for non-persistent patients (N = 25) compared to persistent ones (N = 79) (EUR 5996 and EUR 2448, respectively; data not shown).

The healthcare resources consumption in TAF-treated patients stratified by adherence <80%, 81–95%, and >95% indicated ART prescriptions and specialistic visits/diagnostic tests as the most impactive items (Appendix A). Consistently, during the first year of follow-up, from PDC < 80% to PDC > 95%, the total costs tended to rise, but this effect was mostly due to the higher burden of drug expenses that reflect the growing consumption with increasing adherence (Figure 3). 

Healthcare resources consumptions in non-adherent (PDC < 80%) patients treated with ART other than TAF and TAF-based regimens highlighted that even in non-adherent patients, the majority of consumptions were imputable to ART prescriptions and specialistic visits/diagnostic tests although less markedly in the patients receiving a TAF combination (Appendix A). 

Cost analysis extended to the overall observation period in ART- and TAF-based regimens is shown in Figure 4. The mean annual healthcare direct costs by adherence (PDC > 95%) confirmed that the overall costs tended to be higher in ART-adherent compared to TAF-adherent treatment (EUR 8981 vs. EUR 8523, *p* = 0.070, nearly significant), mostly driven by the expenses for other drugs (EUR 1059 vs. EUR 632, *p* < 0.001) and specialistic visits and outpatient services (EUR 814 vs. EUR 638, *p* = 0.025).

The GLM model showed that the mean yearly non-ART costs increased with older age, specifically by EUR 1364 in patients aged between 51 and 65 years (*p* < 0.001) and EUR 2265 in those over 65 (nearly significant, *p* = 0.079). A worse comorbidity profile was also significantly predictive of higher non-ART costs (EUR 1736 for CCI unit increment, *p* = 0.027) (Appendix A).

## 4. Discussion

The current awareness of the management of HIV patients chronically treated with ART and the subsequent rebounds on healthcare resource consumptions and expenditures is highly variable across the different countries [20,21,22].

The present analysis was conducted in a setting of real-life clinical practice in Italy to describe the state-of-art of HIV-infected patients treated with ART and more specifically with TAF-based regimens, evaluate therapeutic pathways and drug utilization, and give special attention to the impact of adherence, persistence, and therapy discontinuation of TAF-based regimens on HIV management and direct healthcare costs.

The demographic characteristics of the overall ART population reported a male predominance and a mean age of around 49 years, which is in line with other Italian real-world studies on HIV [1,23]. Hypertension and dyslipidemia were among the most frequent comorbidities detected; a French study also observed these two conditions to be highly prevalent in PLWH [24]. We then focused on the 1198 patients prescribed with TAF-based regimen during the inclusion period in order to investigate in this specific group the management of treatments in terms of adherence, persistence, and therapy discontinuation and ultimately how their drug utilization could affect healthcare consumptions and costs. 

There is large body of evidence from national and international studies that, despite the undeniable progression and growing availability of pharmacological options for HIV patients, the goal of maintaining an adequate therapy adherence in the long term is still far from being reached [25]. This point deserves great efforts from the scientific community since adherence to ART therapy has essential repercussions on viral load control and thus patients’ clinical outcomes but also on the consequent savings in terms of healthcare resource consumptions and expenditures for the national health systems [17]. In our study, a higher proportion of adherent patients was found among patients with STR rather than MTR regimens, which is in line with the evidence reported in the literature on higher adherence to STR than MTR [13,26,27]. Interestingly, the comparison between naïve and TAF-switchers showed greater adherence levels in switchers regardless of the cutoff considered (PDC > 95% or PDC > 85%). This is in line with another real-world study in which patients naïve to ART were found less likely to be adherent [28]. Furthermore, a meta-analysis showed that treatment-experienced patients had the highest pooled odds ratio for optimal adherence vs. suboptimal even though this tendency was not statistically significant [29]. Older studies based on first-generation ART suggested that adherence could be unchanged or decreased after change of therapy and that ART switch should trigger intervention to reassess adherence [30], while most recent studies have observed an increase in the number of patients with improved adherence after switching to a second line [31].

Persistence to TAF-based regimens mirrored the trend observed for adherence, with a high proportion of persistent patients among those with STR or switchers. Sutton et al [32] found in a cohort of U.S veterans a proportion of patients persistent to TAF-based regimens ranging from 64–73% for STR and 46–58% for MTR. These values, however, referred to naïve patients, and this could explain the difference with our findings. Indeed, naïve patients had a trend of less persistence than those who were experienced (i.e., switchers to TAF).

To the best of our knowledge, there are limited data on the actual economic burden for patients with TAF-based regimens in Italy, while the literature reports an estimation of cost-effectiveness or budget-impact analysis on such therapies [33,34]. The latter estimated a direct healthcare cost for year 2018 spanning from EUR 8928 to EUR 11,200, which, although lower than our results, is, however, based in real-life settings [33].

In all the scenarios depicted, cost analysis revealed that during the first year of follow-up, the weightiest item on the annual healthcare direct cost per patient was that for ART, which is consistent with previously published data for European countries [35,36,37]. 

The downward trend observed over the last 3 years of inclusion was mainly due to a halving in the costs of other drugs, hospitalizations, and outpatients’ services through the years, suggesting that the introduction of innovative ART on the market could influence the costs for the management of patients. Persistence to TAF-based therapies was associated with significantly lower mean annual costs, in particular for the cost item related to HIV hospitalization. Moreover, the comparison of mean annual healthcare direct costs by adherence (PDC > 85% vs. PDC < 85%) in TAF-based regimen revealed that increased costs not related to ART were observed in non-adherent patients; once again, this trend was evident for HIV hospitalizations. In this regard, Toh et al. [38] explored the economic burden of AIDS-defining illnesses, which are still a major threat for PLWH despite the improvement of life expectancy thanks to ART: the authors found that improved level of adherence is related to an increase in the medical costs but could lead to savings from lower incidence rates of AIDS-defining illnesses and the related costs they carry [38].

Considering the mean annual health care resource consumptions and costs during all available follow-up in TAF-treated patients with respect to adherence, we found tendentially increased consumptions and costs with growing adherence levels but this might be explained by the obviously higher burden of drug expenses in adherent patients, which is in line with the literature [17,38]. Multivariate analysis revealed that older age and more complex comorbidity profile were significant predictors of healthcare costs other than ART, adding to the growing body of evidence that aging is one of the upcoming challenges for HIV management and one of the factors that will contribute to increased HIV care costs [39]. Indeed, PLWH age while taking the life-long ART and therefore will have to address issues related to a multi-comorbidity profile such as polypharmacy, polydoctoring, and iatrogenic diseases [40].

These findings should be interpreted in the light of some limitations related to the observational retrospective design of the analysis, which was based on data extracted from administrative databases. Thus, there might be lacking or insufficient clinical information on a number of other potential confounders (i.e., disease severity, comorbidities, previous virological failures/resistance mutations) that may have influenced the results. Secondly, data on the reasons for changing regimens (switch or discontinuation) are not reported within the database as well as the causes of non-adherence since not-measurable variables as patient attitudes towards medication or social status are not present; similarly, adherence and persistence were evaluated based on the drug dispensed; therefore, the actual use by the patient is unknown. Ultimately, indirect costs or out-of-pocket costs could not be evaluated, as administrative databases contain data on healthcare resources reimbursed by INHS. 

## 5. Conclusions

This real-world analysis provided an in-depth characterization of patients prescribed TAF-based regimens in terms of drug utilization and how this could have affected healthcare resource consumptions and costs. Higher levels of adherence and persistence to TAF-based regimens were detected among users of STR rather than MTR and among experienced patients that switched from an ART- to a TAF-based treatment compared to naïve patients.

A declining trend over the years in healthcare resource consumption and the related direct costs could be suggestive of the changing of costs related to the management of HIV patients due to the advent of novel therapeutic options. The total healthcare costs were mainly driven by ART-related costs, which represent almost 70% of the total expenditure. Patients that showed a higher level of TAF persistence and adherence were reported to have a reduced healthcare cost related to HIV hospitalizations compared to those who were non-persistent or not adherent to TAF-based therapies, suggesting that the optimization of the drug utilization may have a positive impact not only from a clinical point of view but from the perspective of the sustainability of the INHS as well.

## Figures and Tables

**Figure 1 ijerph-20-03789-f001:**
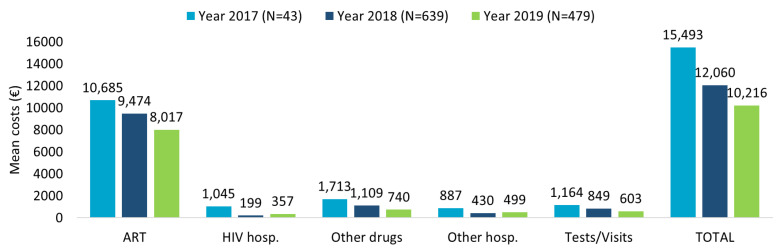
Mean annual health care costs (EUR) in patients with TAF-based regimen during first year of follow-up for the inclusion years 2017–2018–2019.

**Figure 2 ijerph-20-03789-f002:**
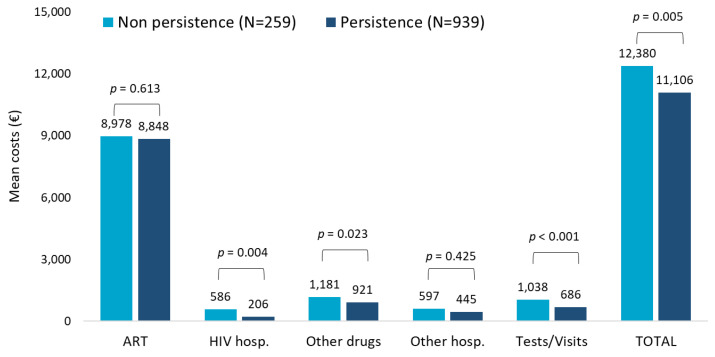
Mean annual health care costs (EUR) during first year of follow-up for patients with TAF-based regimen based on persistence.

**Figure 3 ijerph-20-03789-f003:**
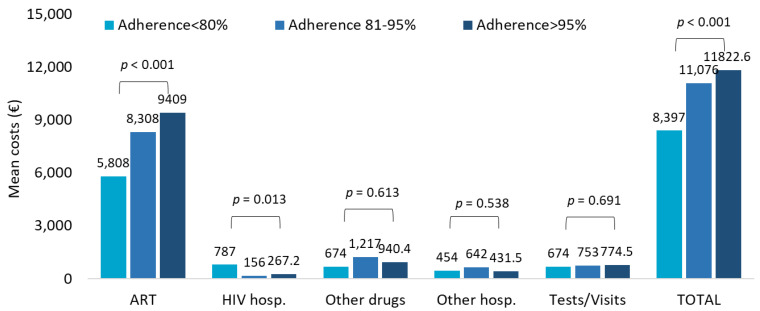
Mean annual health care costs (EUR) during first year of follow-up for patients with TAF-based regimen based on adherence.

**Figure 4 ijerph-20-03789-f004:**
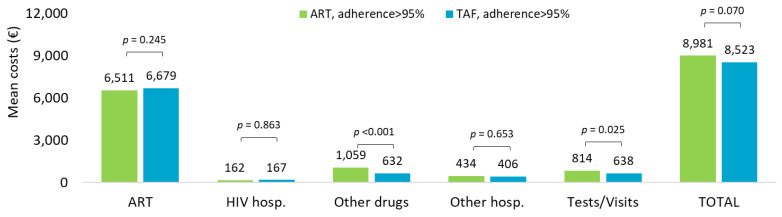
Mean annual health care costs (EUR) during all available follow-up for patients with ART- and TAF-based regimens and based on adherence (PDC > 95%).

**Table 1 ijerph-20-03789-t001:** Demographic and clinical characteristics of the overall ART-treated patients and divided by calendar year. Continuous variables are reported as mean ± standard deviation and categorical variables as numbers and percentages in brackets.

	2015	2016	2017	2018	2019	Overall
Patients, n	966	814	1566	2198	1583	2658
Age, years	48.7 ± 10.2	48.9 ± 10.5	49.5 ± 10.7	50.4 ± 10.7	51.0 ± 10.9	48.6 ± 11.0
Age < 36 years	103 (10.7%)	89 (10.9%)	177 (11.3%)	228 (10.4%)	161 (10.2%)	349 (13.1%)
Age 36–55 years	649 (67.2%)	533 (65.5%)	963 (61.5%)	1303 (59.3%)	860 (54.3%)	1637 (61.6%)
Age > 55 years	214 (22.2%)	192 (23.6%)	426 (27.2%)	667 (30.3%)	562 (35.5%)	672 (25.3%)
Male gender	691 (71.5%)	598 (73.5%)	1129 (72.1%)	1563 (71.1%)	1145 (72.3%)	1903 (71.6%)
Depression	68 (7.0%)	73 (9.0%)	122 (7.8%)	172 (7.8%)	134 (8.5%)	196 (7.4%)
Respiratory disease	152 (15.7%)	127 (15.6%)	351 (22.4%)	397 (18.1%)	212 (13.4%)	422 (15.9%)
Renal failure	8 (0.8%)	14 (1.7%)	27 (1.7%)	35 (1.6%)	18 (1.1%)	35 (1.3%)
Alcohol/drug abuse	30 (3.1%)	30 (3.7%)	55 (3.5%)	99 (4.5%)	73 (4.6%)	104 (3.9%)
Cardiovascular disease	19 (2.0%)	20 (2.5%)	63 (4.0%)	83 (3.8%)	62 (3.9%)	65 (2.4%)
Diabetes	34 (3.5%)	35 (4.3%)	71 (4.5%)	92 (4.2%)	77 (4.9%)	93 (3.5%)
Dyslipidemia	131 (13.6%)	101 (12.4%)	225 (14.4%)	290 (13.2%)	205 (13.0%)	298 (11.2%)
HBV/HCV	71 (7.3%)	73 (9.0%)	154 (9.8%)	196 (8.9%)	136 (8.6%)	207 (7.8%)
Hypertension	178 (18.4%)	178 (21.9%)	355 (22.7%)	524 (23.8%)	399 (25.2%)	519 (19.5%)
Charlson Comorbidity Index	0.2 ± 0.6	0.1 ± 0.5	0.2 ± 0.5	0.2 ± 0.5	0.1 ± 0.5	0.2 ± 0.5
0	840 (87.0%)	737 (90.5%)	1354 (86.5%)	1924 (87.5%)	1418 (89.6%)	2318 (87.2%)
1	96 (9.9%)	60 (7.4%)	172 (11.0%)	220 (10.0%)	137 (8.7%)	275 (10.3%)
≥2	30 (3.1%)	17 (2.1%)	40 (2.6%)	54 (2.5%)	28 (1.8%)	65 (2.4%)

Note: HBV, hepatitis B virus; HCV, hepatitis C virus; NR, not reported (for data privacy when <4 patients are involved).

**Table 2 ijerph-20-03789-t002:** Demographic and clinical characteristics of patients with TAF-based regimen (overall and stratified by MTF/STR and naïve/switchers). Continuous variables are reported as mean ± standard deviation and categorical variables as numbers and percentages in brackets.

Characteristics	TAF-MTR	TAF-STR	Naïve	TAF-Switchers	Overall
Patients, n	514	684	478	720	1198
Age	49.7 ± 9.8	48.0 ± 11.6	47.3 ± 11.5	49.7 ± 10.)	48.7 ± 10.9
Age < 36 years	48 (9.3%)	117 (17.1%)	87 (18.2%)	78 (10.8%)	165 (13.8%)
Age 36–55 years	332 (64.6%)	381 (55.7%)	281 (58.8%)	432 (60.0%)	713 (59.5%)
Age > 55 years	134 (26.1%)	186 (27.2%)	110 (23.0%)	210 (29.2%)	320 (26.7%)
Male	378 (73.5%)	501 (73.2%)	354 (74.1%)	525 (72.9%)	879 (73.4%)
Depression	39 (7.6%)	49 (7.2%)	35 (7.3%)	53 (7.4%)	88 (7.3%)
Respiratory disease	96 (18.7%)	105 (15.4%)	76 (15.9%)	125 (17.4%)	201 (16.8%)
Renal failure	7 (1.4%)	NR	4 (0.8%)	6 (0.8%)	10 (0.8%)
Alcohol/drug abuse	28 (5.4%)	22 (3.2%)	11 (2.3%)	39 (5.4%)	50 (4.2%)
Cardiovascular disease	20 (3.9%)	16 (2.3%)	10 (2.1%)	26 (3.6%)	36 (3.0%)
Diabetes	20 (3.9%)	25 (3.7%)	15 (3.1%)	30 (4.2%)	45 (3.8%)
Dyslipidemia	52 (10.1%)	61 (8.9%)	34 (7.1%)	79 (11.0%)	113 (9.4%)
HBV/HCV	49 (9.5%)	59 (8.6%)	28 (5.9%)	80 (11.1%)	108 (9.0%)
Hypertension	117 (22.8%)	117 (17.1%)	80 (16.7%)	154 (21.4%)	234 (19.5%)
Cancer	28 (5.4%)	16 (2.3%)	11 (2.3%)	33 (4.6%)	44 (3.7%)
Charlson Comorbidity Index	0.2 ± 0.7	0.1 ± 0.4	0.2 ± 0.7	0.1 ± 0.5	0.2 ± 0.6
0	450 (87.5%)	615 (89.9%)	422 (88.3%)	643 (89.3%)	1065 (88.9%)
1	52 (10.1%)	56 (8.2%)	45 (9.4%)	63 (8.8%)	108 (9.0%)
≥2	12 (2.3%)	13 (1.9%)	11 (2.3%)	14 (1.9%)	25 (2.1%)

Note: HBV, hepatitis B virus; HCV, hepatitis C virus; MTR, multi-tablet regimen; NR, not reported (for data privacy when <4 patients are involved); STR, single-tablet regimen; TAF, Tenofovir Alafenamide.

## Data Availability

All data used for the current study are available upon reasonable request to CliCon S.r.l., which is the body entitled to data treatment and analysis by Local Health Units.

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
