# Peer review of "Healthcare Resource Consumption and Related Costs in Patients on Antiretroviral Therapies: Findings from Real-World Data in Italy"

_ijerph, 2023, doi:10.3390/ijerph20053789_

Round 1

Reviewer 1 Report

This is an important area to research and the authors have access to a large database. I have some incomprehensions in terms of methodology:
1) I suggest excluding data on osteoporosis from the analysis due to the lack of data
and patients with this disease,
2)
it is not clear to me how the resource consumption was measured, e.g. in terms of the number of drugs or hospitalizations
3)more information concerning methodology of the recource consumption costs evaluation is needed

Author Response

This is an important area to research and the authors have access to a large database. I have some incomprehensions in terms of methodology:

1) I suggest excluding data on osteoporosis from the analysis due to the lack of data and patients with this disease.

We agree with your comment. The subgroup with osteoporosis was characterized by a very small sample size (below 4 patients), thus results could not be disclosed for data privacy reasons. Any mention of osteoporosis has been now removed in test and tables.

2) it is not clear to me how the resource consumption was measured, e.g. in terms of the number of drugs or hospitalizations

This point has been now better clarified (lines 164-171).

3)more information concerning methodology of the recource consumption costs evaluation is needed

This point has been also better clarified (lines 171-176, 181-184).

Reviewer 2 Report

Line 177 : You have mentioned the GLM models so their is a slight correction for its abbreviation. The word Linea should be replaced as General Linear Model which is the correct way to write full form of GLM models.

Line 397-400 : In these lines the authors have tried to highlight the low cost associated with TAF drug usage. Can you please highlight the reason for non association of HIV patients with its usage even though the past literature states about its cost effectiveness.    

Author Response

Line 177: You have mentioned the GLM models so their is a slight correction for its abbreviation. The word Linea should be replaced as General Linear Model which is the correct way to write full form of GLM models.

This was a typo that we have now corrected (line 187).

Line 397-400 : In these lines the authors have tried to highlight the low cost associated with TAF drug usage. Can you please highlight the reason for non association of HIV patients with its usage even though the past literature states about its cost effectiveness.

In the lines cited by the Reviewer, which we thank for this comment, we wanted to highlight that patients adherent and persistent to TAF had lower HIV related hospitalization rather than patients not persistent or not adherent to TAF-based. We have better clarify the sentence so that it is clear the subjects are the patients (line 407-411). Furthermore, our aim was not to perform a cost-effectiveness analysis, but rather an evaluation of how a proper pharmacoutilzation and therapeutic compliance of ART and in particular TAF-based regimes, can result in cost savings for the Italian NHS. Although ideally, costs should not drive clinical decisions, we are aware that ART are expensive treatments. Anyhow, our analysis was intended to assess whether the efforts to optimize TAF persistence and adherence could have an impact on the sustainability of the NHS.